# Aberrant Methylation of *SLIT2* Gene in Plasma Cell-Free DNA of Non-Small Cell Lung Cancer Patients

**DOI:** 10.3390/cancers14020296

**Published:** 2022-01-07

**Authors:** Yujin Kim, Bo Bin Lee, Dongho Kim, Sang-Won Um, Joungho Han, Young Mog Shim, Duk-Hwan Kim

**Affiliations:** 1Department of Molecular Cell Biology, School of Medicine, Sungkyunkwan University, Suwon 440-746, Korea; yujin0328@hanmail.net (Y.K.); whitebini@hanmail.net (B.B.L.); jindonghao2001@hotmail.com (D.K.); 2Department of Internal Medicine, Samsung Medical Center, School of Medicine, Sungkyunkwan University, Seoul 135-710, Korea; sangwon72.um@samsung.com; 3Department of Pathology, Samsung Medical Center, School of Medicine, Sungkyunkwan University, Seoul 135-710, Korea; joungho.han@samsung.com; 4Department of Thoracic and Cardiovascular Surgery, Samsung Medical Center, School of Medicine, Sungkyunkwan University, Seoul 135-710, Korea; youngmog.shim@samsung.com

**Keywords:** lung cancer, SLIT, methylation, plasma, biomarker

## Abstract

**Simple Summary:**

Despite significant advances in the detection, prevention, and treatment of lung cancer, the prognosis of the patients is still very poor due in part to micrometastasis of cancer cells to surrounding tissues at the time of diagnosis. Therefore, identifying biomarkers for early detection of lung cancer is very important for prolonging the lifespan of patients with lung cancer. The methylation statuses of *SLIT1*, *SLIT2*, *SLIT3* genes were analyzed in bronchial washing, bronchial biopsy, sputum, tumor and matched normal tissues, or plasma samples obtained from a total of 208 non-small cell lung cancer (NSCLC) patients and 121 cancer-free patients to understand the feasibility of the genes as biomarkers for early detection and survival prediction of NSCLC. The present study suggests that aberrant methylation of *SLIT2* in plasma cell-free DNA might be a potential biomarker for the early detection and prognosis prediction of NSCLC patient.

**Abstract:**

This study aimed to understand aberrant methylation of *SLITs* genes as a biomarker for the early detection and prognosis prediction of non-small cell lung cancer (NSCLC). Methylation levels of *SLITs* were determined using the Infinium HumanMethylation450 BeadChip or pyrosequencing. Five CpGs at the CpG island of *SLIT1*, *SLIT2* or *SLIT3* genes were significantly (Bonferroni corrected *p* < 0.05) hypermethylated in tumor tissues obtained from 42 NSCLC patients than in matched normal tissues. Methylation levels of these CpGs did not differ significantly between bronchial washings obtained from 76 NSCLC patients and 60 cancer-free patients. However, methylation levels of *SLIT2* gene were significantly higher in plasma cell-free DNA of 72 NSCLC patients than in that of 61 cancer-free patients (*p* = 0.001, Wilcoxon rank sum test). Prediction of NSCLC using *SLIT2* methylation was achieved with a sensitivity of 73.7% and a specificity of 61.9% in a plasma test dataset (*N* = 40). A Cox proportional hazards model showed that *SLIT2* hypermethylation in plasma cell-free DNA was significantly associated with poor recurrence-free survival (hazards ratio = 2.19, 95% confidence interval = 1.21–4.36, *p* = 0.01). The present study suggests that aberrant methylation of *SLIT2* in plasma cell-free DNA is a valuable biomarker for the early detection of NSCLC and prediction of recurrence-free survival. However, further research is needed with larger sample size to confirm results.

## 1. Introduction

Lung cancer is one of most common causes of cancer-related death worldwide. Despite significant advances in the prevention, detection and treatment of lung cancer over the past 20 years, the prognosis of patients with lung cancer is still extremely poor, with the five-year overall survival rate remaining less than 20% [1]. The prognosis of cancer patients is usually determined by disease stage and treatment modality. Occult metastatic spread of cancer cells from primary tumor to surrounding tissues occurs in more than 50% of lung cancer patients at the time of diagnosis of lung cancer, and it has a very negative effect on patient’s prognosis. The prognosis of patients who undergo curative surgical resection in the early stage of lung cancer is improved with an adjuvant chemotherapy. Thus, identifying biomarkers for early detection of lung cancer and personalized medicine is very important for prolonging the lifespan of lung cancer patients. However, sensitive and specific biomarkers for detecting lung cancer are currently not available in clinical setting.

Mammalian *SLIT1*, *SLIT2* and *SLIT3* genes are located on human chromosome 10q24.1, 4p15.31, and 5q34, respectively. SLITs (SLIT1–3) are highly conserved secreted glycoproteins in vertebrates and mediate biologic effects on cells by interacting with transmembrane receptor ROBOs (ROBO1–4) in many physiological and pathological processes [2]. The SLIT-ROBO pathway is crucial for muscle cell formation, neuronal axon guidance, cell migration, cell proliferation, inflammatory cell chemotaxis and angiogenesis. SLITs and ROBOs are highly expressed in many types of cancer, and the SLIT/ROBO signaling pathway has a positive effect on tumor growth, tumor cell migration, and tumor metastasis [3]. Compared with studies reporting an oncogenic role of SLIT proteins, recent studies have also indicated that they can function as tumor suppressors by inhibiting cancer cell proliferation, cancer progression, and metastasis. The inactivation of SLIT proteins has been reported in several types of cancer, such as breast cancer [4,5,6], colorectal cancer [7,8], gastric cancer [9,10], hepatocellular carcinoma [11,12], lung cancer [13,14,15,16] and thyroid cancer [17].

Hypermethylation of CpGs located at the promoter regions of tumor suppressor genes is now well established as an important mechanism of epigenetic inactivation. Hypermethylation of *SLIT1* gene has been found in colon cancer [18,19] and glioma [20]. However, it has not been reported in lung cancer yet, although hypermethylation of *SLIT2* [21,22,23,24] and *SLIT3* [20,22,24] has been reported in lung cancer.

To understand their feasibility as biomarkers for early detection and survival prediction of non-small cell lung cancer (NSCLC), we analyzed methylation statuses of *SLIT1*, *SLIT2* and *SLIT3* genes using the Infinium HumanMethylation450 BeadChip in 136 bronchial washings, 6 bronchial biopsies, 12 sputums, and 42 lung tumor and matched normal tissues. *SLIT2* methylation was analyzed using pyrosequencing in 133 plasma samples.

## 2. Materials and Methods

### 2.1. Study Population and Samples

A total of 208 NSCLC patients and 121 cancer-free patients who were admitted for diagnosis of suspected lung cancer or for curative surgical resection of lung cancer or who visited for a regular health check-up at Samsung Medical Center in Seoul, Korea between August 1994 and August 2016 participated in this study. The cancer-free patients were recruited from healthy individuals that have not been diagnosed with any cancer till August 2020 or from patients with benign lung diseases such as pneumonia, bronchiolitis, tuberculosis, or actinomycosis. Localized organizing pneumonia and hamartoma were excluded from the study because methylation profiling of both diseases is not well known, which could lead to misclassification. Information related to tissue, bronchial washing, sputum, and bronchial biopsy samples used in this study was described previously [25]. Plasma samples collected after July 2005 were used for analysis of *SLIT2* methylation. To avoid misclassification of case and control, patients with other cancers found during the follow-up for at least 5 years after lung cancer surgery or bronchoscopy or a regular health check-up were excluded from this study. Venous blood was collected into 10 mL Vacutainer^®^ K_3_ EDTA tubes (BD Medical, Franklin Lakes, NJ, USA), kept in a refrigerator and then centrifuged at 1700× *g* for 10 min at 4 °C. The time between sample collection and plasma separation was a minimum of one hour and a maximum of eight hours (Appendix A). Plasma was stored at −80 °C until use. Follow-up after curative resection was conducted by a lung cancer specialized nurse as described previously [26]. This study was approved by the Institutional Review Board of the Samsung Medical Center (IRB#: 2010-07-204). It was conducted in accordance with the Declaration of Helsinki. Written informed consent for the use of pathological specimens was obtained from all participants prior to the procedure. Tumor/node/metastasis (TNM) system provided by The American Joint Committee on Cancer (AJCC) [27] was used for pathologic staging of NSCLC.

### 2.2. Analysis of SLITs Methylation

We have previously analyzed CpGs methylation in 6 bronchial biopsies, 12 sputums, 42 surgically resected tumor and matched normal tissues and 136 bronchial washings obtained from 76 NSCLC patients and 60 cancer-free patients using the Infinium HumanMethylation450 BeadChip [25]. We used the reported data for analyzing aberrant methylation of *SLIT1*, *SLIT2*, and *SLIT3* genes in tumor tissues and bronchial aspirates. Data preprocessing such as batch effect adjustment, background correction, probe filtering and adjustment of performance difference between type I and II probes was performed using the wateRmelon R software package (version 3.1.1). Methylation level (β-value) for each cytosine in a CpG dinucleotide was estimated as the ratio of fluorescent signal from methylated cytosines to the sum of methylated and unmethylated cytosines. CpGs were defined as hypermethylated when β-value was greater than or equal to 0.3 under consideration of background signal.

### 2.3. Feature Selection and Model Building for Lung Cancer Prediction

To select tumor-specific CpGs for lung cancer prediction and model building among differentially methylated CpGs, we randomly divided normal and tumor tissues into training and test datasets at a ratio of 7:3. Supervised machine learning algorithms were applied for feature selection and model building using RapidMiner Studio (version 8.2) in the training dataset. Age-related CpGs and any significantly correlated CpGs in normal or tumor tissues were removed from the model building. The performance of each prediction model was evaluated with test datasets using a receiver operating characteristic (ROC) curve, which was plotted using the statistical software package MedCalc (version 19.0.5).

### 2.4. Analysis of SLIT2 Methylation in Plasma

Methylation levels of cg13281139 locus at the promoter region of *SLIT2* gene were evaluated by pyrosequencing using a PyroMark Q24 ID System (Qiagen, Hilden, Germany) according to the manufacturer’s protocol. DNA was extracted from 2 mL of plasma using the MagMax Cell-Free DNA Isolation Kit (Thermo Fisher Scientific, Waltham, MA, USA) according to the manufacturer’s protocol. The quantity and size of the extracted DNA were measured using an Agilent High Sensitivity DNA Kit (Santa Clara, CA, USA, Part no. 5067-4626). Contaminated genomic DNA was removed using 2% isopropanol. Approximately 20 ng of DNA was treated with bisulfite using an EZ DNA Methylation Gold kit (Zymo Research, Irvine, CA, USA) according to the manufactures’ instructions. Bisulfite-treated DNA was then eluted in 10 μL elution buffer. Primers for pyrosequencing were designed using the PyroMark Assay Design 2.0 software (Qiagen), and primer sequences are listed in Appendix A. *SLIT2* was amplified uisng a PyroMark PCR kit (cat. No 978703; Qiagen) according to the manufacturer’s protocol. Briefly, PCR was performed on a Gene PCR System 9700 (Applied Biosystems, Foster City, CA, USA). PCR was carried out in a final volume of 25 μL comprising 12.5 μL of PyroMark PCR Master Mix (2×), 1 μL of modified DNA, 0.5 μL each of forward and biotinylated reverse primers (0.2 μM final concentration), 2.5 μL of CoralLoad buffer (10×), and 8 μL of RNase-free water. PCR was started with 1 cycle of 95 °C for 15 min, followed by 45 cycles of 95 °C for 15 s, 60 °C for 20 s, 72 °C for 20 s, and 1 cycle of 72 °C for 10 min. PCR products were sequenced on a PyroMark Q24 (Qiagen) using PyroMark Gold Q24 reagents (Qiagen) according to the manufactures’ instructions. To check the accuracy of pyrosequencing, we made 0%, 25%, 50% and 100% methylated samples by mixing methylated (Chemicon, Millipore, Billerica, MA, USA) and unmethylated control DNA (Qiagen) and included them in all experiments. Results were analyzed using the PyroMark Q24 2.0.6 software (Qiagen).

### 2.5. Statistical Analysis

For univariate analysis, the Wilcoxon rank sum test (or *t*-test) and the Fisher’s exact test (or Chi-square test) were used for continuous and categorical variables, respectively. The correlation of methylation levels of selected CpGs was assessed using Spearman’s rank correlation coefficient. The effect of *SLITs* hypermethylation on overall survival or recurrence-free survival (RFS) in lung cancer patients was estimated using the Kaplan-Meier survival curve, and the difference between two survival curves was evaluated by the log-rank test. The hazard ratio of *SLITs* hypermethylation for survival was estimated using Cox proportional hazards regression model after adjusting for potential confounding factors. All statistical analyses were conducted using R software (version 3.6.1).

## 3. Results

### 3.1. Aberrant Methylation of SLIT Genes in Primary Non-Small Cell Lung Cancer

To identify differentially methylated CpGs at the promoter regions of *SLIT* genes, we analyzed methylation statuses of *SLIT1*, *SLIT2* and *SLIT3* genes in tumor and matched normal tissues from 42 NSCLC patients previously reported [25]. Methylation levels of CpGs in tumor tissues were negatively skewed and did not follow a normal distribution (Shapiro-Wilk test, *p* < 0.05). Thus, the Wilcoxon rank sum test was applied to assess the statistical significance. Five CpGs (Figure 1A) with a *p*-value less than or equal to 1.03 × 10 ^−7^ (Bonferroni-corrected threshold) were identified from the 450 K array. Information on these five CpGs is described in Appendix A. Average methylation levels of these five CpGs in tumor tissues were 0.67–0.84. A linear relationship between methylation levels of these CpGs was analyzed in 42 tumor tissues (Figure 1B). Two CpGs (cg13261826 and cg14226472) in *SLIT1* gene were significantly correlated to each other, and a statistically significant correlation was also found between CpGs at the promoter region of *SLIT1* (cg13261826 or cg14226472) and *SLIT3* (cg01163016 or cg26119620). However, a CpG (cg03260566) in *SLIT2* was not correlated to any CpGs in *SLIT1* or *SLIT3*. The association between hypermethylation of CpGs and clinicopathological variables was analyzed. Hypermethylation of the five CpGs was not associated with lymphatic invasion of tumor cells (Wilcoxon rank sum test; Figure 1C). Methylation of cg01663016 (*SLIT3*) was associated with patient age (*r* = 0.40, *p* = 0.009; Figure 1D). Hypermethylation of three CpGs (cg13261825, cg03260566 and cg01663016) in *SLIT1*, *SLIT2* and *SLIT3* genes was found at a higher prevalence in adenocarcinoma than in squamous cell carcinoma, although the difference was not statistically significant (Figure 1E). Methylation levels of the five CpGs were not associated with pack-years, tumor size, pathologic stage, vascular invasion or neural invasion Relationship between hypermethylation of cg03260566 in *SLIT2* and the clinical parameters was shown in Appendix A.

### 3.2. Aberrant Methylation of SLIT Genes in Bronchial Washing, Biopsy, and Sputum Samples

Methylation levels of three CpGs (cg13261825, cg03260566, and cg01663016) were analyzed in 136 bronchial washings (76 NSCLC patients and 60 cancer-free patients), 12 sputum samples and six bronchial biopsy specimens from NSCLC patients to test if those specimens could be used as surrogate samples for assessing aberrant methylation of *SLIT* genes observed in lung tumor tissues. Methylation levels of these three CpGs (Figure 2A–C) in bronchial biopsy, bronchial washing and sputum specimens were significantly lower than those in tumor tissues shown in Figure 1A, but similar to those in normal tissues. In addition, methylation levels of these CpGs in bronchial biopsy samples from NSCLC patients were not significantly different from those in 6 paired bronchial washing samples (*p* > 0.05, Wilcoxon signed-rank test; Figure 2D–F). Based on these observations, sputum, bronchial washing and bronchial biopsy specimens might not be feasible as surrogates for analyzing aberrant methylation of *SLIT* genes in lung tumor tissues.

### 3.3. Prediction of NSCLC Using Methylation Levels of SLIT Genes in Lung Tumor Tissues

To select features and develop a model for predicting NSCLC, we divided 42 lung tumor and matched normal tissues into a training dataset and a test dataset at a ratio of 7 to 3, respectively. The training dataset (*N* = 59) was used for feature selection and model building, and the performance of each model was tested using the test dataset (*N* = 25). Supervised machine learning algorithms such as Naive Bayes (NB), logistic regression, k-nearest neighbor (kNN), support vector machine (SVM), artificial neural network (ANN) and random forest were applied for selecting features and building a model. Correlated CpGs were not included in these models at the same time. Among applied algorithms, a logistic regression model based on cg03260566 in a test dataset (*N* = 25) showed the best performance (sensitivity of 84.2% and specificity of 79.8%) in predicting NSCLC, with the area under the curve (AUC) of 0.76 (95% CI = 0.71–0.92, *p* < 0.0001; Figure 3A). We further evaluated the prediction performance of the same model in TCGA lung cancer data (824 tumor and 74 normal tissues) to understand whether *SLITs* methylation could be applied to other races. Methylation levels of four CpGs in the Cancer Genome Atlas (TCGA) data were similar to those in our data (Appendix A). The TCGA lung cancer data were also divided into a training dataset (*N* = 630) and a test dataset (*N* = 268) for building a model. A logistic regression model based on cg03260566 in the test dataset predicted NSCLC with a sensitivity of 87.2% and a specificity of 78.4% (AUC = 0.83; 95% CI = 0.73–0.92, *p* < 0.0001; Figure 3A). The prediction certainty of NSCLC was found to be high in our data and the TCGA test dataset (Figure 3B).

### 3.4. Aberrant Methylation of SLIT2 in Plasma Cell-Free DNA of Patients with NSCLC

Clinicopathological characteristics of 72 NSCLC patients and 61 cancer-free patients who donated blood samples are described in Appendix A. Genomic DNA (Figure 4A) frequently found in plasma cell-free DNA was removed using 2% isopropanol. The amount of plasma cell-free DNA after isopropanol was very small, and therefore only *SLIT2* methylation was analyzed using pyrosequencing. Pyrosequencing of cg03260566 showing a high performance in predicting NSCLC in tissue samples failed due to an unequal amplification of methylated and unmethylated alleles in PCR reaction. Instead, methylation levels of cg13281139 located approximately 1200 bp upstream from cg03260566 were analyzed. Average methylation levels of cg13281139 were 0.16 and 0.08 in plasmas from 72 NSCLC patients and 61 cancer-free patients, respectively, and the difference was statistically significant irrespective of gender (*p* = 0.001, Wilcoxon rank sum test; Figure 4B). To build a model for lung cancer prediction using methylation levels of cg13281139 in plasma cell-free DNA, the data were divided into a training dataset (70%) and a test dataset (30%). The sensitivity and specificity of a logistic regression model based on cg13281139 in the test dataset (*N* = 40) were 73.7% and 61.9%, respectively (Figure 4C). This finding suggests that *SLIT2* may be a valuable biomarker to detect NSCLC in plasma cell-free DNA.

### 3.5. SLIT2 Hypermethylation in Plasma Cell-Free DNA Is Associated with Poor Recurrence-Free Survival of NSCLC Patients

Clinicopathological significance of *SLIT2* hypermethylation was analyzed in plasma cell-free DNA from 72 NSCLC patients. *SLIT2* was defined as hypermethylated when β-value was greater than or equal to 0.15. *SLIT2* hypermethylation was found in 24 (33%) of 72 NSCLCs. *SLIT2* hypermethylation was not significantly different according to gender (Figure 5A), histology (Figure 5B), or pathologic stage (Figure 5C). In addition, *SLIT2* hypermethylation was not associated with age, smoking status, tumor size, lymphatic invasion, vascular invasion or neural invasion. However, *SLIT2* hypermethylation was significantly associated with tumor recurrence (Figure 5D). The median follow-up period of patients was 5.2 years. Patients with *SLIT2* hypermethylation showed poor recurrence-free survival (RFS) compared to those without (Figure 5E). Five-year recurrence-free survival rates of patients with and without *SLIT2* hypermethylation were 44% and 80%, respectively. Cox proportional hazards analysis showed that RFS of patients with *SLIT2* hypermethylation was 2.19 times (95% CI = 1.21–4.36; *p* = 0.01) poorer than in those without after adjusting for age and pathologic stage (Table 1).

## 4. Discussion

SLIT proteins interact with ROBO receptors, thus playing important roles in cell migration, cell growth, and angiogenesis. Hypermethylation of *SLIT* genes has been reported in diverse types of cancer, including lung cancer. To understand whether hypermethylation of *SLIT* genes can be used as a biomarker for early detection and prognosis prediction of NSCLC, we analyzed methylation statuses of *SLIT* genes in lung cancer tissues, bronchial washing, biopsy and plasma samples. SLIT1 expression is specific to brain and the nervous system [20] and its hypermethylation has not been reported in lung cancer. However, hypermethylation of *SLIT1* in this study was found in 29 (69.0%) of 42 NSCLC tissues. Hypermethylation of *SLIT2* and *SLIT3* occurred in 29 (69.0%) and 23 (54.8%) of 42 NSCLCs, respectively, consistent with previous findings reporting that the frequency of *SLIT3* methylation in lung cancer was less than that of *SLIT2* (20, 22). This suggests that an alternative mechanism other than hypermethylation is responsible for the downregulation of SLIT3. Methylation levels of *SLITs* were compared with those in the TAGA lung cancer data. Similar results were obtained between our data and TCGA lung cancer data (Appendix A). Although the number of normal tissues (*N* = 74) in the TCGA lung cancer data was only about one-tenth of that of tumor tissues (*N* = 824) and the incidence of lung cancer between Koreans and Americans was not exactly the same, the present study suggests that the clinicopatholgocial significance of *SLIT* genes in the present study might be applicable to other races.

Cytologic examination of sputum turned out to be ineffective in enhancing lung cancer detection and reducing lung cancer mortality. To improve their poor efficiency, a number of groups have analyzed methylation changes of multiple tumor suppressor genes in exfoliated bronchial epithelial cells shed from the respiratory tract in addition to sputum. In this study, methylation levels of *SLIT* genes in lung tumor tissues were significantly higher from those in sputum, bronchial washing, and bronchial biopsy samples. In addition, any CpGs in *SLIT* genes on the Illumina Human Methylation 450 K BeadChip did not pass through the Bonferroni corrected significance threshold in exfoliated bronchial epithelial cells obtained through bronchial washing from 76 NSCLC patients and 60 cancer-free patients. Given the likelihood of detecting aberrant methylation in bronchial washing samples depended on tumor location and given that bronchial washing specimens were not good materials for centrally located tumors such as squamous cell carcinoma, we also analyzed methylation changes in bronchial washing samples from squamous cell carcinoma and adenocarcinoma patients separately However, *SLITs* methylation showed no difference between peripherally located adenocarcinomas and cancer-free patients and paired bronchial washing and biopsy samples showed no significant difference in methylation levels of *SLIT* genes either. Based on these results, sputum, bronchial washing and biopsy specimens might not be useful surrogates for analyzing methylation statuses of *SLIT* genes in lung cancer.

Circulating cell-free DNA is a mixture of DNA derived from tumor and normal tissues and is known to contain tumor-associated aberrant methylation in various types of cancer. Analysis of methylation in plasma cell-free DNA through liquid biopsy is considered to have potential for early detection of cancer. Although many epigenetic biomarkers have been discovered for that purpose, it is still challenging to identify biomarkers with clinical accuracy in plasma cell-free DNA. This might be due to the technical difficulty in analyzing highly fragmented cell-free DNA. Among many methods for analyzing CpG methylation in plasma cell-free DNA, pyrosequencing was used in this study. However, we failed to amplify a DNA segment including cg03260566 in plasma cell-free DNA from NSCLC patients. We searched other CpGs around cg03260566 and ultimately analyzed cg13281139, which was approximately 1200 bp away from cg03260566. In tumor tissues, cg13281139 was found to be less frequently hypermethylated than cg03260566 (Appendix A), and methylation status of cg13281139 did not show any relationship to clinical parameters (Appendix A). A significant drawback in this study was the inability to obtain samples from patients’ premalignant disease such as dysplasia and adenoma to analyze its potential as a biomarker for early detection of NSCLC. Therefore, we have compared the methylation levels of cg13281139 in bronchial washing from 60 healthy individuals, matched normal tissues from 42 NSCLC patients, and tumor tissues from stage I–III NSCLC patients (Appendix A). A significant increase in the methylation levels of cg13281139 in stage I compared to normal bronchial washing samples and matched normal tissues suggests the possibility as a biomarker for early detection of NSCLC, but further studies are needed in a large cohort including premalignant lesions. In addition, the sensitivity and specificity of NSCLC prediction using cg13281139 in plasma cell-free DNA were not high (Figure 4C). Failure of methylation analysis in plasma cell-free DNA usually results from primer design failure, uneven amplification of methylated and unmethylated alleles, or amplification failure of fragmented cell-free DNA. These were eventually caused by a lack of knowledge about fragments of cell-free DNA. Internucleosomal linker DNA is preferentially digested by endonuclease during an apoptotic DNA fragmentation [28], and cutting sites in linker DNA are also affected by the density of CpG methylation [29]. To analyze more reliable CpGs in plasma cell-free DNA, it is critical to understand the nucleosomal positioning pattern and ending patterns from plasma cell-free DNA.

In this study, mRNA levels of SLITs and ROBOs were analyzed using data previously obtained from HumanHT-12 expression BeadChips (Illumina, San Diego, CA, USA) in 42 surgically resected tumor and matched normal tissues [30]. mRNA levels of SLIT2, ROBO2, ROBO3, ROBO4 were significantly downregulated in tumor tissues compared to normal tissues (Appendix A). The association between SLIT2 downregulation and patient’s prognosis has been reported in different types of cancer. For example, low expression of SLIT2 is correlated with an upward trend of pathological stage in NSCLC, poor overall and disease-free survival [13]. Knockdown of *SLIT2* increases migration of esophageal squamous cells, and low expression of SLIT2 protein is correlated with poor overall survival and disease-free survival in esophageal squamous cell carcinoma [31]. Expressions of SLIT2 and ROBO1 are downregulated in ductal carcinoma in situ and invasive breast cancer tissues, and such downregulation is associated with poor prognosis and brain metastasis of breast cancer [32]. Survival time for brain glioma patients with decreased *SLIT2* mRNA expression is shorter than for that for those with a normal expression of *SLIT2* [33]. Reduced mRNA expression of *SLIT2* is associated with advanced clinical stage and worse overall survival in diffuse large B cell lymphoma [34]. Papillary thyroid cancers patients with negative SLIT2 expression have significantly increased risk of recurrence and metastasis [17]. SLIT2 mRNA levels in this study showed a negative trend with methylation levels of cg13281139, but were not statistically significant (Appendix A). Nonetheless, the present study suggests that *SLIT2* methylation status in plasma cell-free DNA might be helpful for tracking the recurrence of NSCLC patients. Further studies are needed in a large cohort.

Emerging evidence suggest that SLITs and ROBOs as tumor suppressors act by inhibiting cancer cell proliferation, angiogenesis, invasion, and migration in different types of cancer. Mechanisms of action of SLITs on lung cancer have also been reported by several groups. SLIT2 is frequently inactivated in lung cancer, and knockdown of *SLIT2* decreases the interaction between beta-catenin and E-cadherin/SNAI1 in lung cancer cells, resulting in increased cell migration and reduced cell adhesion [13]. SLIT2 suppresses the migration and invasion of lung cancer cells by regulating RhoA activity [14]. The expression of SLIT3 is downregulated in lung tumor tissues and silencing of *SLIT3* in lung cancer cells induces epithelial-mesenchymal transition by downregulating E-cadherin and enhancing MMP2 and MMP9 expression [35]. In the present study, relationships of *SLITs* methylation with tumor size, pathologic stage, vascular, neural invasion and lymphatic invasion were analyzed, showing no significant relationship. These inconsistent results between in vitro and in vivo studies might be due to the complexity of SLIT/ROBO signaling pathway in tumor proliferation, invasion and metastasis. In this study, the relationship between ROBO expression levels and methylation levels of cg13281139 was analyzed to understand potential cellular mechanisms underlying the negative effect of cg13281139 methylation on RFS. However, no relationship was found between them (Appendix A). Thus, the role of SLITs in clinical lung cancer samples needs to be analyzed by taking into account the expression and/or methylation status of ROBOs and other proteins in the downstream of SLITs.

For analysis of cell-free DNA, it is of the utmost importance to optimize pre-analytical conditions that can affect sample integrity. Several groups reported guidelines for pre-analytical conditions for the analysis of cell-free DNA [36,37]. The interval between blood collection and plasma separation is known to be a factor that can affect the integrity of cell-free DNA. In this study, all plasmas were separated within eight hours of blood collection (Appendix A), and the significant contamination of genomic DNA (Figure 4A) was mostly found in samples whose plasma separation was performed after six hours of blood collection. Accordingly, plasma needs to be isolated as early as possible although blood is stored at 4 °C after collection.

This study was limited by several factors. First, the present study was a retrospective case-control study, which could give rise to a highly biased estimate in population prevalence of NSCLC. Accordingly, prediction performance of a model needs to be validated using a cohort study in different populations. Second, in addition to a small number of total plasma samples, the statistical power was insufficient because the number of normal plasma samples was small compared to that for lung cancer patients. Third, methylation levels of *SLIT1* and *SLIT3* in plasma cell-free DNA were not analyzed due to insufficient amounts of cell-free DNA. *SLIT2* methylation levels were not correlated with *SLIT1* or *SLIT3* methylation. Thus, the addition of *SLIT1* or *SLIT3* methylation level to the present prediction model might improve the prediction performance of NSCLC. Fourth, the mechanism of action of SLIT2 in inhibiting recurrence of NSCLC was not fully elucidated. Fifth, there was significant genomic DNA contamination in plasma (Figure 4A) due to the long interval between blood collection and plasma separation. The genomic DNA was properly removed using 2% isopropanol, but loss of cell-free DNA was also significant (Appendix A). Sixth, there was a possibility that degradation of genomic DNA from lymphocytes results in small fragments that may increase the amount of plasma cell-free DNA. However, we could not separate small DNA fragments of lymphocytes from cell-free DNA technically.

In conclusion, the present study suggests that aberrant methylation of *SLIT2* gene in plasma cell-free DNA might be a potential biomarker for detecting NSCLC and predicting recurrence-free survival. Further research is needed with larger sample size to confirm results.

## Figures and Tables

**Figure 1 cancers-14-00296-f001:**
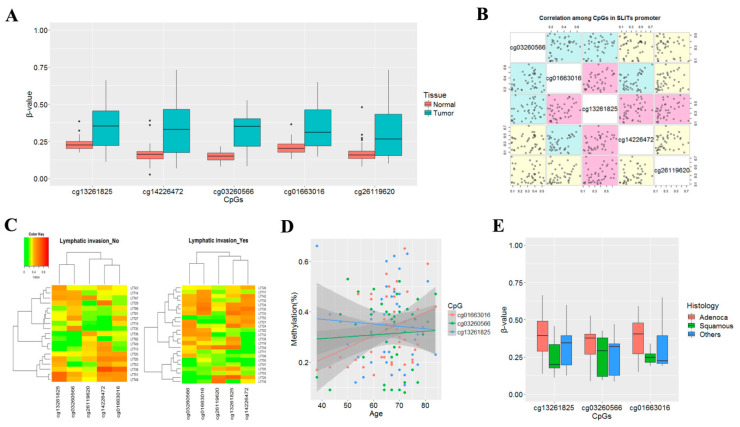
Relationship between aberrant methylation of *SLIT* genes and clinicopathological features in 42 patients with NSCLC. (**A**) Methylation levels of five CpGs at the CpG island regions of *SLIT1*, *SLIT2*, and *SLIT3* genes were compared between tumor and matched normal tissues from 42 NSCLC patients. Y-axis indicates β-value. (**B**) Correlation among methylation levels of five CpGs was analyzed for 42 tumor tissues using Spearman’s correlation coefficients. Magenta color indicates *p* < 0.05. (**C**–**E**) Methylation levels of CpGs at *SLIT1*, *SLIT2* and *SLIT3* genes were compared according to lymphatic invasion (**C**), age (**D**), and histologic subtypes (**E**).

**Figure 2 cancers-14-00296-f002:**
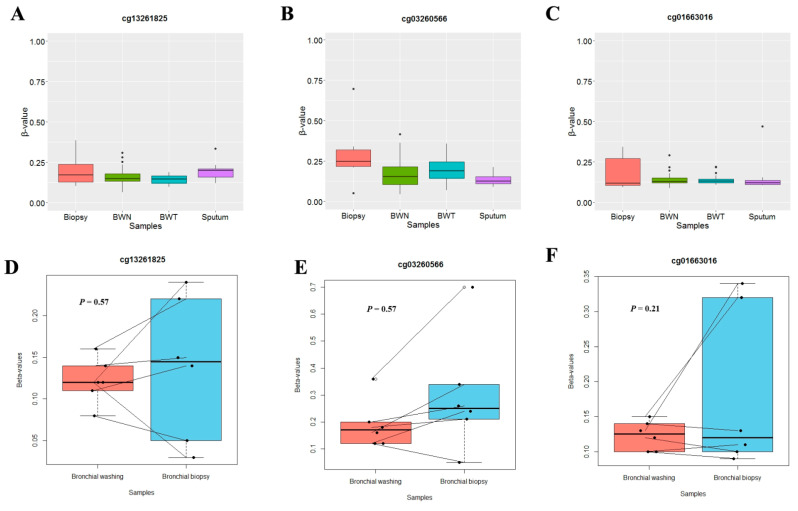
Aberrant methylation of *SLIT* genes in bronchial washing, biopsy, and sputum samples. (**A**–**C**) Methylation levels of three CpGs ((**A**) cg13261825, (**B**) cg03260566 and (**C**) cg01663016) were compared in six bronchial biopsy and 12 sputum samples from NSCLC patients and in bronchial washing samples from 60 cancer-free patients (BWN) and 76 NSCLC patients (BWT). (**D**–**F**) Methylation levels of these three CpGs were also analyzed between paired bronchial washing and bronchial biopsy specimens from six NSCLC patients. *p*-values are based on Wilcoxon signed rank test. Y-axis indicates β-value.

**Figure 3 cancers-14-00296-f003:**
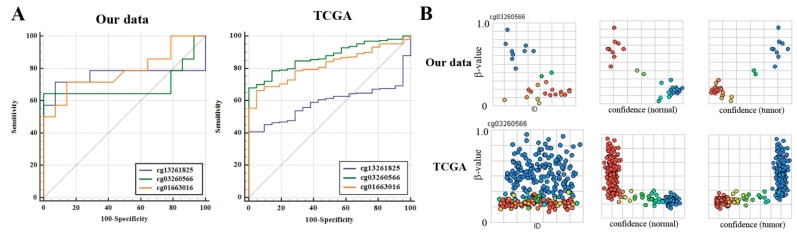
Evaluation of prediction performance of a model in our data and TCGA lung cancer data. (**A**) Prediction performance of a logistic regression model based on three CpGs was evaluated in our data (left) and TCGA (right) test dataset. ROC curves were plotted using the MedCalc software. (**B**) Prediction certainty of a logistic regression model based on cg03260566 was estimated in a test dataset. Y-axis indicates β-value. “ID” in the left panel represents identification number of test samples. In the middle and right panels, the X-axis indicates confidence (normal or tumor) that predicts samples as normal (middle panel) or tumor (right panel) for each β-value on the Y-axis. The sky blue and red orange circles indicate high and low confidence, respectively.

**Figure 4 cancers-14-00296-f004:**
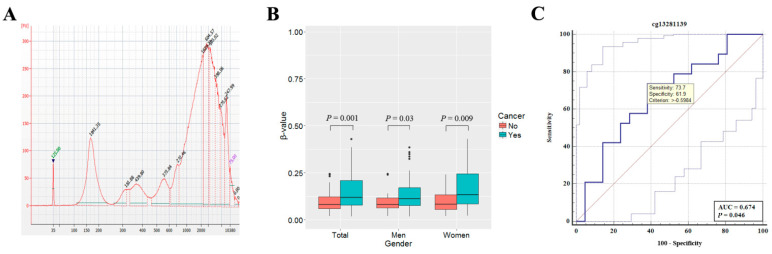
Analysis of *SLIT2* methylation in plasma cell-free DNA of NSCLC patients. (**A**) Cell-free DNA extracted from plasma was analyzed using an Agilent DNA high sensitivity Kit on an Agilent Bioanalyzer 2100 instrument. Electropherogram shows representative sample with contaminated genomic DNA fragments longer than 400 bp. (**B**) Methylation levels of *SLIT2* (cg13281139) in plasma cell-free DNA were compared between 72 NSCLC patients and 61 cancer-free patients according to gender. *p*-values were based on Wilcoxon rank sum test. (**C**) True positive (sensitivity) and false positive (1-specificity) rates of a logistic regression model based on cg13281139 in plasma cell-free DNA were evaluated with a test dataset (*N* = 40). ROC curve with 95% confidence interval was plotted using the MedCalc software (version 19.0.5).

**Figure 5 cancers-14-00296-f005:**
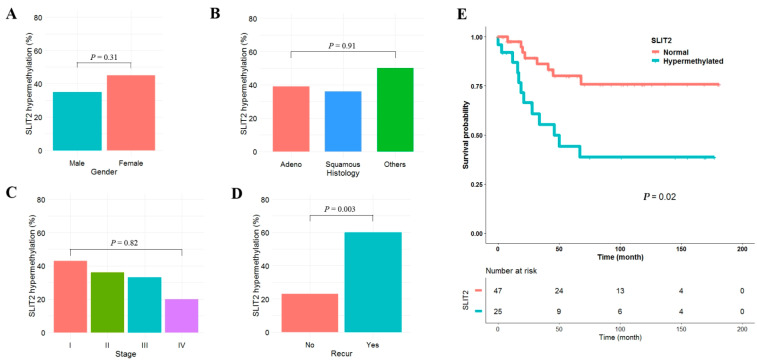
Negative effect of *SLIT2* hypermethylation in plasma cell-free DNA on recurrence-free survival of NSCLC patients. (**A**–**D**) Methylation levels of *SLIT2* in plasma cell-free DNA from 72 NSCLC patients were compared according to gender (**A**), histology (**B**), pathologic stage (**C**), and recurrence (**D**). “Adeno” and “Squamous” indicate “adenocarcinoma” and “squamous cell carcinoma”, respectively. *p*-values were based on Fisher’s exact test or Chi-square test. (**E**) The relationship between methylation level of *SLIT2* in plasma cell-free DNA and recurrence-free survival was analyzed using Kaplan-Meier survival estimates, and *p*-value was calculated using the log-rank test.

**Table 1 cancers-14-00296-t001:** Cox proportional hazards analysis of survival according to *SLIT2* hypermethylation.

Survival	SLIT2 Hypermethylation	HR	95% CI	*p*-Value
Overall survival	No	1.00	-	-
	Yes	1.84	0.76–5.11	0.38
Recurrence-free survival	No	1.00	-	-
	Yes	2.19	1.21–4.36	0.01 ^1^

Abbreviations: HR, hazard ratio; CI, confidence interval. ^1^ Adjusted for age and pathologic stage.

## Data Availability

The data presented in this study are available in the article and Appendix A.

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
