# Peer review of "Aberrant Methylation of SLIT2 Gene in Plasma Cell-Free DNA of Non-Small Cell Lung Cancer Patients"

_cancers, 2022, doi:10.3390/cancers14020296_

Round 1

Reviewer 1 Report

The work is interesting but needs further investigation. The authors described the methylation status of the SLIT1, 2, 3 genes using previously collected data, but then only methylation of the SLIT2 gene in plasma was examined and not in the same region. One criticism is that the pre-analytical phase of blood collection and plasma treatment is not well described. One paper in this area is cited here as an example: J Mol Diagn. 2018 Nov;20(6):883-892. doi: 10.1016/j.jmoldx.2018.07.005. Epub 2018 Aug 28.PMID: 30165204. In any case, the time between sample collection and plasma separation must be reported, as this is crucial for the integrity of the ccfDNA. In addition, the authors must specify the centrifugation force as rcf instead of rpm. This latter parameter depends on the rotor diameter. This point is also important because the fact that the authors had to frequently clear the plasma sample of genomic DNA with isopropanol could indicate poor quality plasma preparation for ccfDNA extraction.

In addition, part of this study was derived from previous data on the methylation status of genes in tissues, but the hypermethylation of the selected region of SLIT2 did not correlate with the clinicopathological stage of NSCLC. Moreover, the analysis of methylation of SLIT2 in plasma failed for the region analyzed in tissues (cg03260566). The authors thus demonstrated that a different region of SLIT2 (cg13281139) was hypermethylated in tumors compared to non-tumor controls. The authors need to analyze the methylation status of the same region (cg13281139) in tumor and normal tissues and possibly mRNA expression in selected samples. Finally, the possible role of hypermethylation of the cg13281139 region as a biomarker for early NSCLC is not entirely conclusive in my opinion. The authors need to provide a more detailed explanation based on their available data that can support this hypothesis

Reviewer 2 Report

In the present manuscript, Kim et al have analyzed methylation statuses of SLIT genes in samples from 76 patients with lung cancer using biopsies, bronchial washing, biopsy, and plasma samples. The main finding of the study was the detection of higher methylation levels of the SLIT2 gene in plasma cell-free DNA of 72 NSCLC patients compared to that of 61 cancer-free patients.  Of note, SLIT2 hypermethylation in plasma cell-free DNA was significantly associated with poor recurrence-free survival. The authors suggest that the aberrant methylation of the SLIT2 gene in plasma cell-free DNA might be a potential biomarker in NSCLC. Overall, the manuscript was well-written, and the quality of the Figures and Tables is adequate. The experiments included controls and were described in sufficient detail. The authors have preemptively discussed their study limitations such as sample size and failure to amplify DNA segments including cg03260566 in plasma cell-free DNA from NSCLC patients. Therefore, the authors' conclusions are well supported by the results. Nevertheless, the authors should consider the major criticism detailed below which would strengthen their argument.

  1. To shed some light on the potential cellular mechanism involved in the association between the hypermethylation of the SLIT2 gene and the shorter recurrence free-survival, it would be relevant to check ROBO expression levels in the tissue.

Minor criticism

  1. The results concerning the association between SLIT2 hypermethylation with age, smoking status, tumor size, lymphatic invasion, vascular invasion, or neural invasion should be included in the main text.

Round 2

Reviewer 1 Report

The authors have given the time between blood collection and plasma processing and declared this to be the average time. This is not appropriate and they need to change the time between blood collection and plasma processing from a minimum ... to a maximum ... ... The average time can also be added as median and standard deviation, but not as the only information. Furthermore, the authors introduce in lines 430-433: "Fifth, there was significant genomic DNA contamination in plasma (Fig. 4A) due to the long interval between blood collection and plasma separation (36). The genomic DNA was properly removed using 2% isopropanol, but loss of cell-free DNA was also significant (Supplementary Fig. S6)". Can the authors rule out the possibility that degradation of genomic DNA from lymphocytes results in small fragments that may increase the amount of cfDNA (in this context, figure S6 must include both electropherograms before and after 2% isopropanol precipitation)? The authors need to discuss this point appropriately or, alternatively, demonstrate similar results in a small sample of plasma specimens where the pre-analytical conditions are suitable for obtaining cfDNA from blood without contamination by genomic DNA from lymphocytes.
